



# Effects of Ozone Levels on Climate Through Earth History

Russell Deitrick[1] and Colin Goldblatt[1]

[1]School of Earth and Ocean Sciences, University of Victoria, Victoria, British Columbia, Canada

**Correspondence:** Russell Deitrick (rdeitrick@uvic.ca)

**Abstract.** Molecular oxygen in our atmosphere has increased from less than a part per million in the Archean Eon, to a fraction of a percent in the Proterozoic, and finally to modern levels during the Phanerozoic. While oxygen itself has only minor radiative and climatic effects, the accompanying ozone has important consequences for Earth climate. Using the Community Earth System Model (CESM), a 3-D general circulation model, we test the effects of various levels of ozone on Earth's climate. When $CO_2$ is held constant, the global mean surface temperature decreases with decreasing ozone, with a maximum drop of $\sim 3.5$ K at near total ozone removal. By supplementing our GCM results with 1-D radiative flux calculations, we are able to test which changes to the atmosphere are responsible for this temperature change. We find that the surface temperature change is caused mostly by the stratosphere being much colder when ozone is absent; this makes it drier, substantially weakening the greenhouse effect. We also examine the effect of the structure of the upper troposphere and lower stratosphere on the formation of clouds, and on the global circulation. At low ozone, both high and low clouds become more abundant, due to changes in the tropospheric stability. These generate opposing short-wave and long-wave radiative forcings that are nearly equal. The Hadley circulation and tropospheric jet streams are strengthened, while the stratospheric polar jets are weakened, the latter being a direct consequence of the change in stratospheric temperatures. This work identifies the major climatic impacts of ozone, an important piece of the evolution of Earth's atmosphere.

## 1 Introduction

The ozone levels in Earth's atmosphere have changed dramatically throughout the planet's history. While the focus of many studies of ozone is its protective influence on biota, ozone also has an effect on climate. In this work, we use climate models to systematically study how ozone affects temperatures at the surface and throughout the atmosphere, in addition to other climatic responses.

Ozone has several contrasting radiative effects. It absorbs strongly in the UV around 0.2 to 0.3 $\mu$m (see, e.g., Petty, 2006). This absorption (of solar UV light) occurs primarily in the stratosphere, where ozone abundance is greatest, and causes the temperature inversion there. The warmer upper atmosphere weakens the greenhouse effect provided by water and $CO_2$, because warmer gases emit more radiation to space. In addition, ozone absorbs thermal radiation from Earth's surface and lower atmosphere around 9.6 $\mu$m, and thus it contributes its own greenhouse effect. Further, by altering the vertical temperature structure, ozone alters the abundances of water, in both its condensed and vapor forms. In turn, this affects climate through the greenhouse effect of water vapor and clouds, and the albedo (reflectivity) of clouds.



Ozone is produced in Earth's atmosphere through photochemistry: molecular oxygen is broken up by solar UV photons shortward of 0.2 $\mu$m, after which the atomic oxygen reacts with molecular oxygen to form ozone, or $O_3$ (Chapman, 1930). Therefore, the history of ozone is intimately connected to the history of molecular oxygen. For the first $\sim 2$ billion years

of Earth's history (the Hadean and Archean Eons), oxygen and ozone were present only in abundances of $\lesssim 10^{-7}$ parts-per-volume (Zahnle et al., 2006; Lyons et al., 2014; Catling and Zahnle, 2020). The Archean ended roughly 2.4 billion years ago with the largest chemical change the atmosphere has ever experienced: the great oxidation event (GOE), during which molecular oxygen rose to levels of $\sim$0.001 to 0.01 (Zahnle et al., 2006; Lyons et al., 2014; Catling and Zahnle, 2020). The ozone layer, which today provides a protective shield from harmful UV radiation, first formed during this event (Ratner and

Walker, 1972; Walker, 1978a, b), though it was probably weaker and lower in the atmosphere (Levine et al., 1979; Kasting and Donahue, 1980; Garduño Ruiz et al., 2022). During the Proterozoic, the oxygen and ozone levels stayed broadly within these levels (Lyons et al., 2014). A second rise of oxygen occurred at the end of the Proterozoic Eon and beginning of the Phanerozoic Eon, bringing oxygen up to its modern value of $\sim 21\%$ (Lyons et al., 2014). At this time, the ozone column became thicker and the peak abundance moved upward to create the familiar structure of the stratosphere (Levine et al., 1979;

Kasting and Donahue, 1980; Garduño Ruiz et al., 2022).

The photochemical production of ozone as a product of oxygen has been extensively studied using 1-D models (Ratner and Walker, 1972; Levine et al., 1979; Kasting and Donahue, 1980; Goldblatt et al., 2006; Gregory et al., 2021; Wogan et al., 2022; Garduño Ruiz et al., 2022) and very recently in 3-D general circulation models (GCMs) (Way et al., 2017; Cooke et al., 2022; Jaziri et al., 2022). A key take-away from these studies is that ozone abundances are non-linear in oxygen concentrations—for

example, ozone reaches near modern levels even at $O_2$ levels of $\lesssim 10^{-3}$ (Garduño Ruiz et al., 2022). This, combined with the staircase-like increase in $O_2$, means that $O_3$ developed in a non-linear fashion in time.

In addition to the protection it affords life, the ozone layer has climatic impacts. For modern Earth, depletions in ozone due to anthropogenic activities are thought to affect the atmospheric and oceanic circulation, particularly in the Southern hemisphere (Keeble et al., 2014; Solomon et al., 2015; Seviour et al., 2017a, b; Bais et al., 2019). Prior modeling work on the larger

extremes of the distant past (i.e., the Archean and Proterozoic Eons) offers contradictory results. Such extremes have been principally studied using 1-D models (Morss and Kuhn, 1978; Levine and Boughner, 1979; Visconti, 1982; Kasting, 1987; Francois and Gerard, 1988). These works found that the global mean surface temperatures decreased by 5-7 K when ozone was removed. In contrast to the 1-D models, Jenkins (1995), and Jenkins (1999) found, using a 3-D GCM, that a reduction of ozone raised global mean surface temperatures by $\sim 2$ K. This was explained as a result of an increase in cloud top altitude,

which in turn led to an increased greenhouse effect, since higher, colder clouds do not radiate as well. Another result from a GCM found that the zonal jets increased in strength as ozone was reduced (Kiehl and Boville, 1988). This work did not explore the temperature response, though the zonal flow is undoubtedly connected to temperature. More recently, the Archean climate has been studied with GCMs in Wolf and Toon (2013); Charnay et al. (2013), though these studies were not designed to isolate the impact of depleted ozone. The effects of these extreme changes of ozone in the past remain under-studied, especially with

more up-to-date models.





Changes in atmospheric oxidation are intimately connected with the so-called faint young sun paradox (FYSP) (Feulner, 2012). The theory of stellar evolution tells us that the sun was dimmer in the past, around 80% of the present day level at the time of the GOE (Gough, 1981; Bahcall et al., 2001). Yet, except for short-lived global (or nearly global) glaciations around the time of the GOE (Kirschvink, 1992), surface temperatures of the distant past were similar to modern levels (Sagan and
Mullen, 1972; Zahnle et al., 2007; Feulner, 2012; Catling and Zahnle, 2020; Charnay et al., 2020). A handful of solutions to this apparent paradox have been proposed, most notably an increased level of greenhouse gases such as $CO_2$ and methane ($CH_4$) (Feulner, 2012), or a cocktail of reduced greenhouse gases (Byrne and Goldblatt, 2014b). The fainter sun can be at least partially offset due to a decrease in stratocumulus cloud formation, and therefore planetary albedo (Goldblatt et al., 2021). The formation of an ozone layer and the resulting changes in cloud formation may play a role as well. The effect due to ozone
(directly and indirectly through clouds and other processes) has not been fully quantified.

The goal of the present manuscript is to begin the process of quantifying the role of ozone on global climate at levels spanning the GOE to present day. As our first foray into this topic, we present results from a general circulation model (GCM) of present-day Earth with large changes in the ozone levels. This allows us to isolate the effect of ozone and prepare for future studies which will include the effects of a reduced solar constant, large amounts of methane, and an increased rotation rate.
Section 2 describes our model setup and execution. Section 3 details the results from our 3-D GCM runs and 1-D radiative transfer calculations. Finally, Section 4 outlines the key findings and places them in the context of past works.

## 2 Methods

### 2.1 General circulation model

To test the effects of ozone on the global climate, we use the Community Earth System Model (CESM) version 1.2.2[1]. CESM
is a general circulation model (GCM) developed and maintained by the National Center for Atmospheric Research (NCAR) in Boulder, CO, for studying the modern Earth's atmosphere. The component set used for all simulations is "E_1850_CAM5": pre-industrial Earth with the Community Atmosphere Model (CAM) version 5.0 atmosphere model, Community Land Model (CLM) version 4.0, Community Ice CodE (CICE) version 4.0, and the River Transport Model (RTM). We also use the Data Ocean Model (DOCN) in the "slab ocean model" (SOM) mode[2]. The model grid is "f19_g16", which uses finite-volumes that
are $1.9° \times 2.5°$ in latitude and longitude for CAM and CLM. The ocean and sea ice models use a displaced pole grid (wherein the location of the pole is moved away from the rotation axis to circumvent numerical issues) with $1°$ elements. The river model uses $0.5°$ elements along the river paths. There are thirty vertical levels in hybrid sigma-pressure coordinates (Collins et al., 2004), spanning from the surface ($\sim 1000$ hPa) to $\sim 2$ hPa. All simulations are run for 60 years, which is sufficient to eliminate the spin-up phase (usually 5-10 years). We then average years 31-60 to eliminate variability. Note that the use of a 3-D dynamic

---

[1]Available here: https://www.cesm.ucar.edu/models/cesm1.2/

[2]We use the slab ocean data file `pop_frc.b.e11.B1850C5CN.f09_g16.005.082914.nc`, which is available here: https://svn-ccsm-inputdata. cgd.ucar.edu/trunk/inputdata/ocn/docn7/SOM/



ocean would require a significantly longer integration time in order to reach steady-state, because of the long circulation time scale of the ocean. The slab ocean does not necessitate such a long time scale, though it is naturally less realistic.

In our simulations (Table 1), we alter only two quantities from the baseline component set: ozone and carbon dioxide. For ozone, we use vertical profiles from the photo-chemical calculations of Garduño Ruiz et al. (2022). In all cases, the ozone is horizontally uniform and constant in time. This is done both to simplify the analysis and because 3-D chemical calculations are not available for the full range of oxygen and ozone. For the present atmospheric level (PAL) reference case with constant $O_3$, we compare first to a simulation[3] with the default, horizontally- and seasonally-varying ozone profiles. The latter case, with varying $O_3$, has ozone levels given in monthly averages in a prescribed input file[4] at each latitude and longitude. The ozone distributions in this file are derived from Dütsch (1978) and Chervin (1986). At each point in time, CAM then interpolates between the nearest monthly averages to compute the current ozone abundance. The two PAL simulations are otherwise identical. In all our constant $O_3$ cases, we customize the ozone by modifying the input ozone file prior to running the GCM. We commonly refer to each GCM by the near-surface amount of molecular oxygen corresponding to each ozone profile (the "nominal $O_2$" level), however, we do not modify the molecular oxygen abundance in the GCM as this is expected to have a minimal impact on climate, assuming the surface pressure is held at 1 bar.

The oxygen and ozone profiles are shown in Figure 1. It is worth noting that as the oxygen is decreased, the peak concentration of ozone occurs lower in the atmosphere, at higher pressures, due to the lower optical depth of molecular oxygen in the stratosphere. This effect has important consequences for cold-trapping of water vapor, as the absorption of solar photons by ozone peaks lower in the atmosphere, warming the tropopause in some cases (see Section 3.3).

We run two "sequences" of simulations. In the Constant $CO_2$ sequence, all simulations have the same $CO_2$ level. In the Temperature Control sequence, we adjust the $CO_2$ level to achieve a roughly constant mean surface temperature, based on the final surface temperature of the Constant $CO_2$ simulations. There is only one simulation with a nominal $O_2$ level of $10^{-2}$ as this simulation has a similar surface temperature to the PAL case at the same $CO_2$ level. This simulation is listed in both sequences in Table 1.

We estimate the $CO_2$ concentration necessary to induce a temperature change using the following equation from Byrne and Goldblatt (2014a),

$$\mathcal{F} = 5.32\ln\left(C/C_0\right) + 0.39\left[\ln\left(C/C_0\right)\right]^2, \tag{1}$$

where $C$ is the concentration of $CO_2$, $C_0 = 2.78 \times 10^{-4}$, and $\mathcal{F}$ is the resulting forcing. The surface temperature and forcing are approximately related by

$$\Delta T = T(C) - T(2.847 \times 10^{-4}) \tag{2}$$
$$\approx c\Delta\mathcal{F} = c\left[\mathcal{F}(C) - \mathcal{F}(2.847 \times 10^{-4})\right], \tag{3}$$

where $c$ is a proportionality constant equal to 1 K m$^2$ W$^{-1}$. The resulting values are listed in Table 1. As we see in Section 3.3, this estimate is quite good at reproducing the desired surface temperature in the GCM.

---

[3]Previously run by Brandon Smith, another member of our research group
[4]Titled `ozone_1.9x2.5_L26_1850clim_c090420.nc`





**Table 1.** GCM Simulations

| Simulation type | Representative eon | Nominal $O_2$ mixing ratio$^\star$ at surface (mol mol$^{-1}$) | $CO_2$ mixing ratio (mol mol$^{-1}$) | Notes |
|---|---|---|---|---|
| **Reference** | | | | |
| | Phanerozoic | 0.21 (PAL) | $2.85 \times 10^{-4}$ | Constant $O_3$ |
| | Phanerozoic | 0.21 (PAL) | $2.85 \times 10^{-4}$ | Varying $O_3$ |
| **Temperature Control** | | | | |
| | Proterozoic | $10^{-2}$ | $2.85 \times 10^{-4}$ | Same as Constant $CO_2$ |
| | Proterozoic | $10^{-3}$ | $3.02 \times 10^{-4}$ | - |
| | Proterozoic | $10^{-4}$ | $4.27 \times 10^{-4}$ | - |
| | Archean | $10^{-7}$ | $5.08 \times 10^{-4}$ | - |
| | Archean | $10^{-9}$ | $4.98 \times 10^{-4}$ | - |
| **Constant $CO_2$** | | | | |
| | Proterozoic | $10^{-2}$ | $2.85 \times 10^{-4}$ | Same as Temperature Control |
| | Proterozoic | $10^{-3}$ | $2.85 \times 10^{-4}$ | - |
| | Proterozoic | $10^{-4}$ | $2.85 \times 10^{-4}$ | - |
| | Archean | $10^{-7}$ | $2.85 \times 10^{-4}$ | - |
| | Archean | $10^{-9}$ | $2.85 \times 10^{-4}$ | - |

$^\star$Determines the $O_3$ mixing ratio, however, the $O_2$ abundance itself is always set to the default value in our CESM simulations. Except for the PAL cases, the values listed are rounded to the nearest order of magnitude.

The solar constant, rotation rate, and land-ocean distribution are all kept at the modern configuration for the purposes of this study, though we acknowledge that each one of these will have a significant impact on climate. In particular, the larger solar constant of the present day (compared to 2.4 Ga) means that we must use a much smaller greenhouse gas abundance than appropriate for the time of the GOE (Catling and Zahnle, 2020; Charnay et al., 2020). Thus the climate sensitivity of our simulations is possibly larger than the reality (see, for example, Sect. 4.2 of Charnay et al., 2020). Nonetheless, we can gain insight about the relative strength of forcings due to various processes.

We use two quantities diagnostic of low cloud formation over marine environments: the lower tropospheric stability (LTS) and estimated inversion strength (EIS). The LTS is simply the difference in potential temperature between the surface and the 700 hPa level. For the EIS, we use Equation 4 of Wood and Bretherton (2006). Their formula is dependent on the height of the lifting condensation level (LCL); for this we use the approximation by Lawrence (2005). The LTS and EIS are averaged over ocean regions only. In regions where the boundary layer inversion does not exist, the calculation of the EIS may produce negative values. Thus, we additionally exclude negative values from the average. The LTS and EIS are first calculated for the monthly-averaged data, then averaged over the last thirty years of integration.





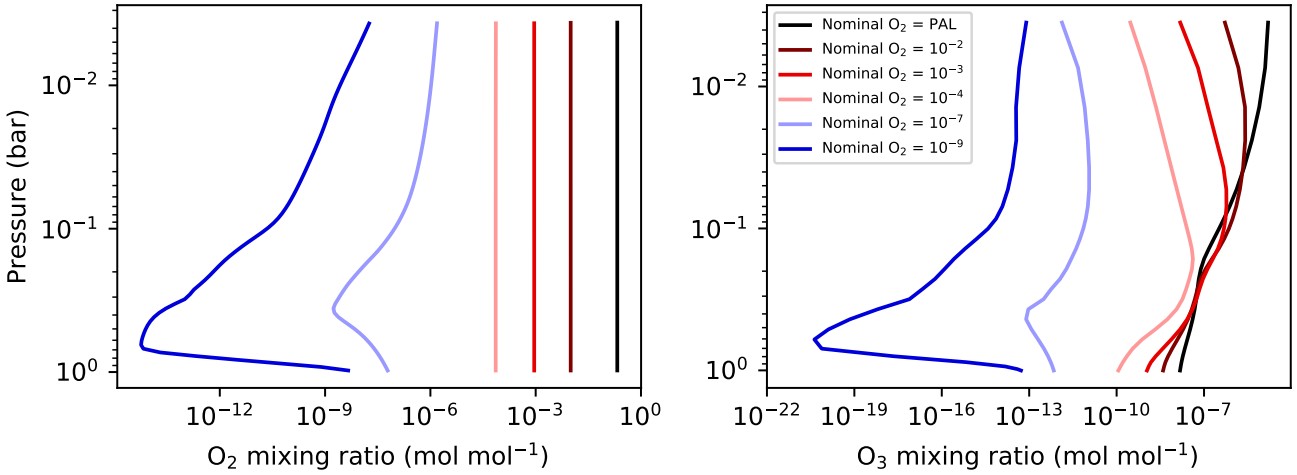

**Figure 1.** Vertical profiles of molecular oxygen ($O_2$) and resulting ozone ($O_3$) from the photochemical models of Garduño Ruiz et al. (2022). The latter are used as input to the CESM simulations listed in Table 1. We do not vary the $O_2$ in the GCM simulations, but frequently refer to the simulations by the near-surface $O_2$ mixing ratio as this is a more intuitive quantity than the $O_3$ mixing ratio.

The cloud top is useful for understanding the cloud radiative forcing, so we estimate the temperature and pressure level of the tops of ice clouds. This is computed by estimating the highest level at which the cloud mass mixing ratio exceeds some arbitrary threshold. Here, we choose a threshold of $10^{-8}$ kg kg$^{-1}$, which does well at distinguishing the different cases.

### 2.2  1-D radiative transfer model

We use a 1-D radiative transfer (RT) model for sensitivity experiments examining the effect of each of the changes involving
ozone. These are: ozone itself, molecular oxygen, and humidity. We do not vary clouds in the 1-D experiments because cloud forcings are provided by the 3-D GCM.

For this, we use the RRTMG code (Mlawer et al., 1997; Clough et al., 2005; Iacono et al., 2008); specifically, we use the Python implementation inside the package climt (Monteiro et al., 2018). The comparison to the CESM results should be quite good as the CAM module uses RRTMG internally for its radiation, though here we configure the model in slightly different
ways.

For the humidity and temperature profiles, we use the global mean profiles from each GCM run, averaged over the last 30 years of integration. For ozone and oxygen, we use the profiles from Garduño Ruiz et al. (2022) corresponding to the ozone profiles used in each GCM run. Note again that oxygen is not varied in GCM simulations themselves. We vary oxygen to test the validity of our assumption that $O_2$ would have negligible effect on the climate through radiation.
We use RRTMG to calculate the short-wave and long-wave fluxes of the constant $O_3$ PAL case, which we use as our reference case. We then "perturb" one variable at a time by swapping the PAL profile for one from another GCM simulation.





For example, in one perturbed case, we use the humidity and oxygen from the PAL simulation, plus the ozone column from Pre-GOE 1. This particular case allows us to isolate the forcing due to the change in ozone.

In all cases, the surface pressure is kept the same, which means that the background $N_2$ increases when the $O_2$ abundance is reduced. $CO_2$ is kept at 284.7 ppmv in all calculations.

For clouds, we use the values for cloud fraction, cloud water path, and cloud particle size from Table 3 of Goldblatt and Zahnle (2011). To handle cloud overlap, we also follow the method of that study, which requires computation of eight columns. From the eight columns, we compute a weighted average, where the weights are determined by the cloud fraction in each layer. While RRTMG contains a built in method for modelling cloud overlap (Pincus et al., 2003; Oreopoulos et al., 2012), this method is randomized, and thus many repeated calculations are necessary to produce a consistent result. This is valid within a GCM (and is indeed used within CAM), where the integration over time will tend to average out stochastic noise and produce smooth, repeatable results, but for 1-D calculations it becomes more burdensome than the eight column method of Goldblatt and Zahnle (2011).

We define the forcing as

$$\mathcal{F} = F'_{\text{net}} - F_{\text{net}}, \tag{4}$$

where $F_{\text{net}}$ is the net flux from the PAL calculation, and $F'_{\text{net}}$ is the net flux from the perturbed state (in which one profile is replaced by that from a different GCM simulation). We define the net flux as

$$F_{\text{net}} = F_\downarrow - F_\uparrow, \tag{5}$$

where $F_\downarrow$ and $F_\uparrow$ are the down-welling and up-welling fluxes, respectively, for either the short-wave, long-wave, or combined total (SW plus LW). The sign convention used here means that positive forcing corresponds to heating of the surface and atmosphere with respect to the reference case; negative forcing then corresponds to cooling with respect to the reference case.

We compare the forcing taken at approximately the 200 hPa level and the top-of-atmosphere (TOA) level. The former roughly represents the tropopause, though the location of the tropopause varies across the temperature profiles. Further, this assumption ignores horizontal and seasonal variations in the tropopause location within the GCM simulation. Nevertheless, forcings calculated at nearby layers are all similar to those calculated at 200 hPa, so the result is insensitive to the exact location of the tropopause. Forcings calculated at the tropopause are indicative of the energy budget of the troposphere and surface, while those calculated at the TOA are sensitive to the energy budget of the atmosphere as a whole.

## 3 Results

### 3.1 Nomenclature

Throughout the analysis, we refer to "high" and "low" clouds, as well as "short-wave" and "long-wave" radiation. Therefore, a few definitions are in order. First, CAM separates clouds into three categories based on pressure level:

- Low: 700 mbar $< p <$ 1200 mbar



- – Mid: 400 mbar $< p <$ 700 mbar

- – High: 50 mbar $< p <$ 400 mbar

Where we present or discuss cloud fractions or densities in these categories, they are vertically integrated over the relevant pressure layers.

Additionally, fluxes and forcings are separated into two categories based on wavelength:

- – Short-wave (SW): 0.2 $\mu$m $< \lambda <$ 12.2 $\mu$m

- – Long-wave (LW): 3.1 $\mu$m $< \lambda <$ 1000 $\mu$m

Note that the two wavelength ranges overlap, though the SW flux *long*-ward of $\sim 3$ $\mu$m and the LW flux *short*-ward of $\sim 3$ $\mu$m are negligible. In reality, it is perhaps more accurate to refer to the two regimes as "solar" and "thermal" radiation, rather than short-wave and long-wave, which are treated differently and independently inside the model. Cloud forcing is defined as the difference between true net flux (with clouds) and that of the same column under clear skies. The clear sky fluxes and TOA forcings are built-in calculations provided in CAM. Net fluxes are defined with positive oriented downward, so that positive

forcing has a warming effect on the atmosphere and negative forcing has a cooling effect.

### 3.2    Validation of approximations

We begin our presentation of results by validating our constant ozone approximation. We do this by comparing global mean quantities in our PAL simulations with constant ozone and seasonally-, horizontally-varying ozone profiles (the two Reference simulations in Table 1). We take the global mean vertical profiles for temperature, specific humidity, relative humidity, cloud

fraction, ice cloud density and liquid cloud density, all averaged over years 31-60. The maximum differences in temperature, relative humidity and cloud fraction between the two simulations are $\sim 0.1$ K, 0.3%, 0.001, respectively. For specific humidity and cloud density, we take a difference of the logarithm of each quantity (or a log of the ratio). For specific humidity, ice cloud density, and liquid cloud density, these are 0.008, 0.7, and 0.19, respectively. As can be seen in Figure 3, the profiles for the two simulations are indistinguishable by eye. The most significant difference occurs in the ice cloud density, where the ratio of the

two reaches a factor of $\sim 10^{0.7} \sim 5$. However, this occurs at high altitudes where the values of ice cloud density are very small, $\sim 10^{-22}$ kg m$^{-3}$. Everywhere else, the difference in log of the ice cloud density is $\lesssim 0.15$. We conclude that our treatment of ozone as constant in time and horizontally uniform is valid for the purposes of this study.

We further demonstrate that leaving molecular oxygen at its modern level in the GCM has a negligible impact, assuming the atmosphere is always 1 bar. We do this by calculating the forcing, relative to PAL concentrations, incurred by changing $O_2$ in

RRTMG. The forcing that results from changing the $O_2$ mixing ratio to $\sim 10^{-9}$ mol mol$^{-1}$ (our lowest ozone case) is $-0.97$ W m$^{-2}$ at the top-of-atmosphere and $-0.18$ W m$^{-2}$ at the tropopause. These are an order-of-magnitude less than the effects of ozone, humidity, and clouds. Therefore, we conclude that changing oxygen would have negligible consequences in the GCM.



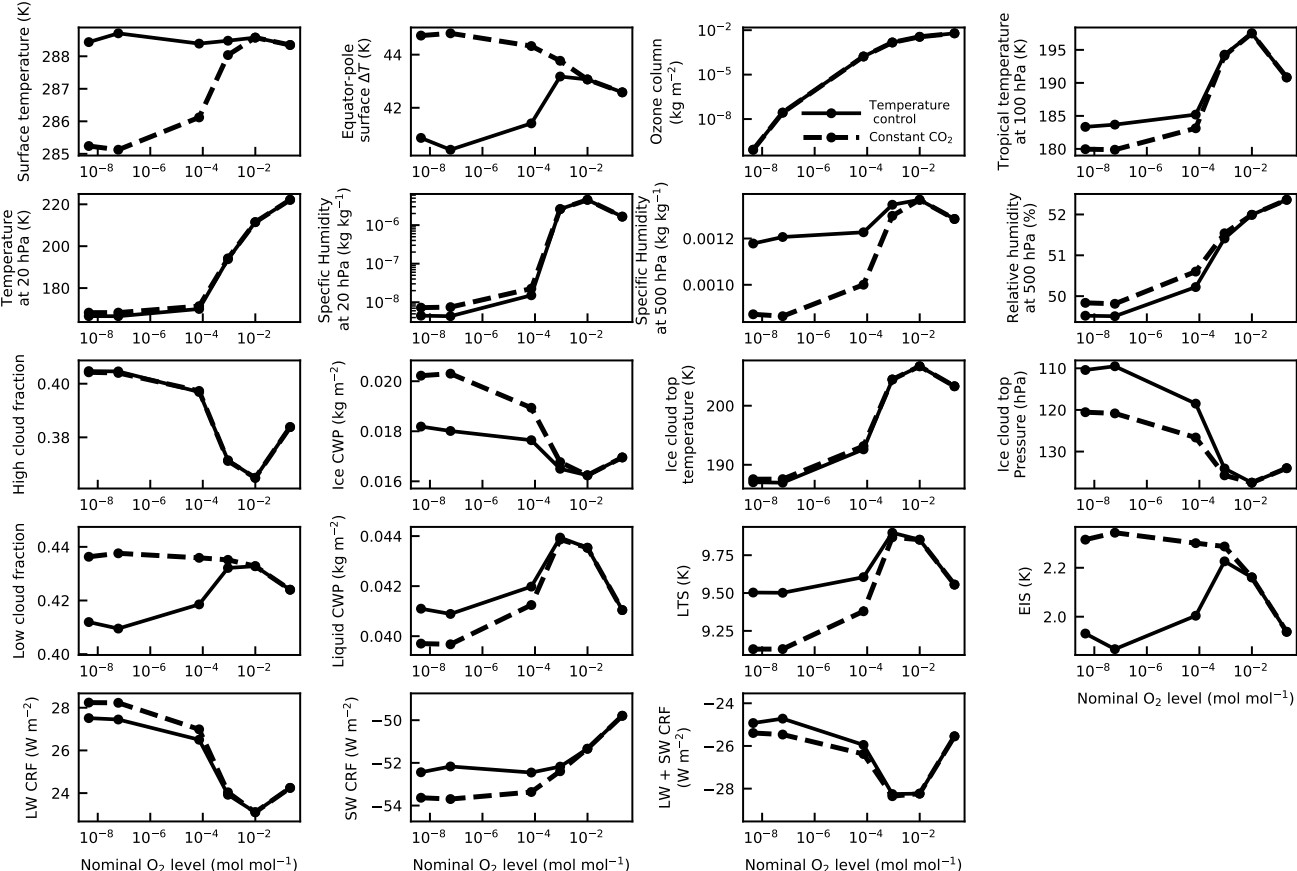

**Figure 2.** Output from the GCM simulations as a function of nominal $O_2$ level. First row: global mean surface temperatures, equator-to-pole surface temperature difference (the difference in the average between latitudes -15° to 15° and latitudes > 60°), the integrated ozone column, and the tropical tropopause ($p \sim 100$ hPa) temperature (averaged from latitudes -15° to 15°). Second row: global mean temperature at 20 hPa, global mean specific humidity at 20 hPa, at 500 hPa, and global mean relative humidity at 500 hPa. Third row: high cloud fraction (50 hPa $< p <$ 400 hPa), integrated ice cloud water path, estimated ice cloud top temperature, and estimated ice cloud top pressure, all globally averaged. Fourth row: low cloud fraction ($p >$ 700 hPa), integrated liquid cloud water path, lower tropospheric stability, and estimated (boundary layer) inversion strength. The former three are globally averaged, the LTS is averaged over marine locations, and the EIS is averaged over marine locations while also excluding negative values. Fifth row: long-wave cloud radiative forcing, short-wave cloud radiative forcing, and the sum of the LW and SW cloud forcing.




**Figure 3.** Globally averaged vertical profiles for all simulations. From left to right and top to bottom, these are temperature, cloud fraction, specific humidity, ice cloud density, relative humidity, and liquid water (cloud) density. Solid lines correspond to the Temperature Control simulations, dashed lines to the Constant $CO_2$ simulations. The insets in the left columm focus on the region with $p > 0.5$ bar. Note that the pressure ranges in the right hand plots are different from the left.






**Figure 4.** Zonally-averaged temperature, specific humidity, zonal velocity, and relative humidity for the Temperature Control and Reference GCM simulations. The top row shows fields from the PAL case. The rows below show the difference fields for each lower ozone case with respect to the PAL case.





### 3.3 Thermal response to low ozone

When $CO_2$ is held constant, the global mean surface temperature tends to decrease with decreasing ozone (Figure 2). The
exception is the Proterozoic case at a nominal $O_2$ level of $10^{-2}$, which is the same temperature to within the numerical noise of
the simulations. From nominal $O_2 = 10^{-3}$ to $10^{-4}$, there is a $\sim 2$ K drop in surface temperature. Moving to nominal $O_2 = 10^{-7}$
there is a further $\sim 1$ K decrease, and essentially no change at lower $O_2$. However, the most dramatic effect takes place in the
stratosphere.

Stratospheric temperatures decrease by as much as $\sim 80$ K for the lowest ozone cases (Figures 3 and 4)—a direct result of
the decreased UV absorption by ozone at high altitudes. This holds for both Temperature Control and Constant $CO_2$ cases.
Additionally, the three lowest ozone cases (nominal $O_2 = 10^{-4}$, $10^{-7}$, and $10^{-9}$) are significantly drier in the stratosphere, due
to the colder temperatures. Figures 2 and 3 display this lowered specific humidity in the global mean and Figure 4 shows this in
the zonal mean. Conversely, the cases at nominal $O_2 = 10^{-2}$ and $10^{-3}$ have higher specific humidity than the PAL case in the
stratosphere despite the colder temperatures. This is a consequence of a warmer tropical tropopause (Figures 2 and 4), which
allows deep convection to loft a greater amount of moisture into the stratosphere. The warmer tropopause, in these cases, is
itself a result of the peak in ozone abundance occurring lower in the atmosphere than in the PAL case (Figure 1).

The decrease in surface temperature at low ozone (in the Constant $CO_2$ simulations) is a consequence of its effect on
the stratosphere. The causal chain is as follows: lowering ozone decreases stratospheric temperatures, which then decreases
stratospheric humidity. The stratosphere then imparts a negative forcing on the troposphere and surface by diminishing the
greenhouse effect due to water vapor. The decrease in ozone itself generates a positive forcing because of the increase short-
wave flux reaching the surface. This is not enough, however, to fully compensate for the effect of decreased stratospheric
humidity. The negative forcing due to stratospheric humidity changes can be seen in Figure 6. The forcing at the tropopause
due to humidity reaches $\sim -2.5$ W m$^{-2}$ in the lowest ozone cases. The tropopause forcing due to ozone is $\sim 1$ W m$^{-2}$ in
those same cases (Figure 5). Adding the two results in a net forcing of $\sim -1.5$ W m$^{-2}$, which by itself would lower surface
temperatures by 1.5 K, given the roughly 1 K W$^{-1}$ m$^2$ sensitivity. This is not enough to explain the entire temperature change
felt by the surface, but positive feedbacks amplify the response.

Exploring the forcings in more detail, we start with that due to ozone (Figure 5). Both upward and downward short-wave
fluxes increase at all levels as ozone decreases. In the downward beam, this can be interpreted simply as less absorption. In the
upward beam, the decreased absorption allows more solar photons to be scattered (via Rayleigh scattering), which increases the
upward flux as well. At lower ozone, the net SW flux at the top-of-atmosphere (TOA) is decreased, compared to the PAL case,
resulting in a negative forcing overall. At the tropopause, however, the net SW flux is greater than in the PAL case, resulting
in a positive forcing. The difference between TOA forcing and tropopause forcing gives rise to the temperature decrease in
the stratosphere. Less energy from the solar radiation is absorbed by the whole atmosphere, but more of it is reaching the
troposphere. There is an increase in the upward long-wave flux as well, because ozone absorbs around 9.6 $\mu$m. This results in
a decrease in net LW flux at all pressure levels compared to the PAL case and therefore a negative forcing at all levels. As a



**Figure 5.** Fluxes and forcings resulting from single-column sensitivity experiments with RRTMG, changing the entire ozone column. The ozone profiles correspond to the cases indicated in the legend, while all other profiles take on the PAL values. From top to bottom, these are the short-wave fluxes, the difference in short-wave fluxes between each case and the reference (black curves), the long-wave fluxes, the difference in long-wave fluxes between each case and the reference, and the forcings as a function of nominal $O_2$ level, at the top-of-atmosphere (TOA) and the 200 hPa level (roughly the tropopause). The first and third rows contain insets zoomed on the regions below 200 hPa.





greenhouse gas, removal of ozone cools the atmosphere. Combined, the SW forcing dominates and the total forcing ends up at $1.5$ W m$^{-2}$ in the lowest ozone cases.

The forcing for stratospheric humidity (Figure 6) is more straightforward as it is dominated by the long-wave. In fact, the TOA forcing due to stratospheric humidity is minimal in both LW and SW. Instead, we see the greenhouse effect of stratospheric

water vapor from the forcing at the tropopause—the downward long-wave fluxes are increased for the cases with increased humidity (nominal O$_2 = 10^{-2}$ and $10^{-3}$) and decreased for the cases with decreased humidity (nominal O$_2 = 10^{-4}$–$10^{-9}$).

In combination, changing ozone and stratospheric humidity has a net cooling effect in the lowest ozone cases and a net warming in the cases nearest to the PAL (Figure 7). The net SW flux at the tropopause is increased in all lower ozone cases, while the net LW flux at the tropopause varies more with the ozone abundance. In the nominal O$_2 = 10^9$–$10^{-4}$ cases, the

decrease LW flux outweighs the increase in SW, leading to a negative forcing. In the cases with nominal O$_2 = 10^{-3}$ and $10^{-2}$, the net LW flux is nearly zero at the tropopause, thus the increase in SW flux leads to a positive forcing. This positive forcing of $\sim 2$ W m$^{-2}$ in these cases does not result in an increase in surface temperature in the GCM because of the increase in negative cloud forcing (Figure 2). The forcing at the TOA is negative and dominated by ozone, in all cases, as evident by comparing with Figure 5.

**3.4 Cloud response to low ozone**

One of the most obvious effects of lowering ozone is an increase in high clouds. This holds whether we hold CO$_2$ or surface temperature constant, and results in an increase in the greenhouse effect from clouds. Low clouds, however, are more sensitive to our assumptions (Constant CO$_2$ vs. Temperature Control). In the lowest ozone cases, the net changes in SW forcing and LW forcing are nearly equal and opposite, indicating a net change in cloud forcing close to zero, compared to the PAL case.

The three lowest ozone cases have increased long-wave cloud forcing compared with the PAL case (Figure 2) in both the Temperature Control and Constant CO$_2$ sequences, while the cases with nominal O$_2 = 10^{-2}$ and $10^{-3}$ have a slight decrease. These are a consequence of the change in cloud fraction, ice cloud water path, and the estimated ice cloud top temperature, which all affect the LW fluxes. In the three lowest ozone, Constant CO$_2$ cases, the LW cloud forcing is slightly larger than in the corresponding Temperature Control cases due to the increased cloud water path. The cloud top pressure, in the same

cases, is also higher than the Temperature Control, which correlates with the lower tropical tropopause temperatures. A more effective cold trap prevents the clouds from reaching as high in atmosphere. However, the dominant effects of ozone on high clouds are insensitive to the surface and troposphere temperatures, as illustrated by comparing the Temperature Control and Constant CO$_2$ curves in the LW cloud forcing.

In the three lowest ozone cases, high clouds are denser, taller, and cover more area at lower ozone (Figures 3 and 9). This

correlates roughly, but not perfectly, with the relative humidity (Figures 2-4)—ice clouds extend higher in the atmosphere and relative humidity is higher in the stratosphere, owing to the cooler temperatures.

The effects on the short-wave cloud forcing and low clouds are *slightly* more sensitive to our assumptions (controlling CO$_2$ abundance vs. surface temperature), as we see in the lower panels of Figure 2. In general, the SW cloud forcing increases (becomes more negative) as ozone is decreased. Thus, in the lowest ozone cases, the net cloud forcing is nearly identical to



**Figure 6.** Fluxes and forcings resulting from single-column sensitivity experiments with RRTMG, changing the stratospheric humidity. The stratospheric humidity profiles correspond to the cases indicated in the legend, while all other profiles take on the PAL values. From top to bottom, these are the short-wave fluxes, the difference in short-wave fluxes between each case and the reference (black curves), the long-wave fluxes, the difference in long-wave fluxes between each case and the reference, and the forcings as a function of nominal $O_2$ level, at the top-of-atmosphere (TOA) and the 200 hPa level (roughly the tropopause). The first and third rows contain insets zoomed on the regions below 200 hPa.


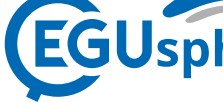

**Figure 7.** Fluxes and forcings resulting from single-column sensitivity experiments with RRTMG, changing the ozone column and stratospheric humidity together. The ozone and stratospheric humidity profiles correspond to the cases indicated in the legend, while all other profiles take on the PAL values. From top to bottom, these are the short-wave fluxes, the difference in short-wave fluxes between each case and the reference (black curves), the long-wave fluxes, the difference in long-wave fluxes between each case and the reference, and the forcings as a function of nominal $O_2$ level, at the top-of-atmosphere (TOA) and the 200 hPa level (roughly the tropopause). The first and third rows contain insets zoomed on the regions below 200 hPa.





the PAL case. The low cloud fraction increases monotonically with decreasing ozone in the Constant $CO_2$ cases, but follows a trend that mirrors that of high clouds in the Temperature Control cases. While low clouds are probably the largest contributor to the SW cloud forcing, the mid- and high-clouds must also be significant contributors. The lower tropospheric stability (LTS) and estimated inversion strength (EIS) have both been used as predictors of low marine clouds. Comparing the Temperature Control and Constant $CO_2$ sequences, LTS appears to be the better predictor of liquid cloud water path, while EIS is the better

predictor of low cloud fraction.

Comparing the Temperature control and Constant $CO_2$ experiments, we can distinguish effects that result from changes to stratospheric temperatures and changes to surface or tropospheric temperatures. An important caveat of this analysis: it is not a perfect separation because the effect of ozone is more latitudinally-dependent than $CO_2$. Hence, lowering ozone while increasing $CO_2$ decreases the equator-to-pole temperature gradient (Figure 2). So although this exchange does keep

global mean surface temperatures roughly constant, the temperature distribution of the surface and troposphere do change significantly, which has regional consequences for cloud formation.

A regional picture of cloud fractions and forcings is presented in Figure 8, for the Temperature Control sequence. As expected LW cloud forcing is strongly correlated with high cloud fraction; similarly, SW cloud forcing is anti-correlated with low cloud fraction, though not as strongly (e.g., forcing is relatively weak in polar regions, where low cloud fraction is highest).

In general, the increase in high clouds with decreasing ozone is clearly evident here. High cloud fraction is highest over the tropical oceans; high clouds increase at low ozone most strongly in these regions, suggesting that this increase is dominated by cumulonimbus anvil cloud formation. Basically, the weakened stability of the upper troposphere and lower stratosphere allows for more vigorous convection. For low clouds, the trend with ozone is less obvious, in part because the latitudinal temperature distribution is changing with $CO_2$.

## 3.5 Response of circulation to low ozone

Circulation is affected by a decrease in ozone, as well. This can be seen in the zonal wind (Figure 4) and the mass stream function (Figure 9). As ozone decreases, the high altitude polar jets weakens dramatically, while the mid-latitude jets grow in strength. This is particularly true in the southern hemisphere. Winds in both regions are likely close to geostrophic balance. The change in wind speeds is then a result of the difference in horizontal temperature gradient between cases, via the thermal wind

equation (see, for example, Holton and Hakim, 2012). We can confirm this by integrating the thermal wind equation upward from the tropopause, using the zonal- and time-averaged temperature field. For the zonal component, this is

$$\frac{\partial u}{\partial \ln p} = -\frac{R}{f}\frac{\partial T}{\partial y}, \tag{6}$$

where $u$ is the zonal velocity, $p$ is the pressure, $R$ is the gas constant, $f$ is the Coriolis parameter, $T$ is the temperature, and $y$ is the meridional coordinate in meters. In Figure 10, we show the zonal winds for all Temperature Control cases. In the

same figure, we compare the wind field determined by Equation 6. While this does overestimate the peak wind speeds in the stratosphere, it does remarkably well at reproducing the wind structure and the trend from high to low ozone. Thus the changes in stratospheric zonal flow with ozone are largely a direct of the temperature changes.



**Figure 8.** Horizontal maps of the LW cloud forcing, high cloud fraction, SW cloud forcing, and low cloud fraction for the Reference PAL case, as well as the difference between each of the low ozone, Temperature Control cases and the PAL case.





**Figure 9.** Zonally-averaged mass stream function, cloud fraction, ice cloud density (in log units), and liquid cloud density (also in log units) for the Temperature Control and Reference GCM simulations. The top row shows fields from the PAL case. In the stream function and cloud fraction columns, the rows below show the difference fields for each lower ozone case with respect to the PAL case. In the cloud density columns, the rows below show the absolute fields for each lower ozone case.




Lowering ozone increases the strength of meridional circulation in the southern hemisphere (i.e., the Hadley and Ferrel cells circulate a greater amount of air), while circulation in the northern hemisphere is less affected. Figure 9 shows the Eulerian mean stream function, a diagnostic of the meridional circulation (Pauluis et al., 2008). From nominal $O_2 = $ PAL to $10^{-9}$, the southern Hadley cell increases from $\sim 9.7 \times 10^{10}$ kg s$^{-1}$ to $\sim 11.7 \times 10^{10}$ kg s$^{-1}$; the northern increases from $\sim 8.9 \times 10^{10}$ kg s$^{-1}$ to $\sim 9.6 \times 10^{10}$ kg s$^{-1}$, though it weakens slightly in the nominal $O_2 = 10^{-2}$ and $10^{-3}$ cases. The increase in strength at low ozone is likely related to the decreased opacity of the atmosphere in both short-wave and long-wave: with less ozone, the SW fluxes reaching the surface and the LW fluxes emerging from the troposphere both increase. This generates more energy to power thermally-direct overturning by allowing more heating at the equatorial surface and more efficient radiative cooling along the descending branch of the cell. The Ferrel cells and tropospheric jets increase in strength alongside the Hadley cells, as these are connected via continuity and the thermal wind.

## 4   Discussion and conclusions

Here, we have revisited the climatic impact of low ozone, with a focus on the climate before and after the great oxidation event. We ran the CESM general circulation model with modern conditions, varying the amount of ozone and $CO_2$. Several past works have examined the climatic impact of complete ozone removal (Kasting, 1987; Francois and Gerard, 1988; Jenkins, 1995, 1999). Our nominal $O_2 = 10^{-9}$ cases have ozone low enough to be analogous to these prior works. While Kasting (1987) and Francois and Gerard (1988) found cooler surface temperatures, using 1-D radiative-convective models, the 3-D GCM modelling of Jenkins (1995, 1999) resulted in a *warmer* global mean surface temperature, using the GENESIS model. Now, with CESM 1.2.2, our results (for Constant $CO_2$) are more similar to the Kasting (1987) and Francois and Gerard (1988) results, albeit with a smaller global-mean surface temperature difference: a cooling of $\sim 3.5$ K, compared to their $\sim 5$ K and $\sim 7$ K, respectively.

Jenkins (1995) and Jenkins (1999) pointed to increased LW forcing to explain the warmer climate produced without ozone. The explanation is sensible—weaker stratification in the upper atmosphere allows for the enhanced formation of cirrus clouds due to enhanced convection and higher relative humidity. We find a similar increase in high clouds in our three lowest ozone simulations, however, this is accompanied by a decrease in the global-mean surface temperature. In our case, the change in LW cloud forcing is accompanied by an opposing change in SW cloud forcing that nearly cancels it. Instead, we find that the surface temperature is more affected by stratospheric humidity, with a drier stratosphere leading to a cooler surface.

Decreasing ozone in isolation lowers the global-mean surface temperature by $\lesssim 3.5$ K. Given the similarity of the simulations with nominal $O_2 = 10^{-7}$ and $10^{-9}$, it is unlikely that the complete removal of ozone would result in a temperature change greater than this. Because of the diminished heating in the stratosphere, low ozone cases have decreased specific humidity and, at the same time, increased relative humidity in the upper troposphere and more high clouds. This increases the overall long-wave cloud forcing, but also decreases the greenhouse effect due to water vapor. A small increase in low clouds also strengthens the (negative) short-wave cloud forcing.






**Figure 10.** Zonal-mean zonal wind for the Temperature Control and Reference GCM simulations. The left column shows the output from the GCM. The right shows the approximation given by integrating the thermal wind equation from the tropopause upward, using the zonal-mean temperature field. The gray box indicates the troposphere, where we instead show the GCM output (i.e., it is identical to the left column) because this region is not as well represented by the thermal wind equation.





Ultimately, however, it is the lowered greenhouse at low humidity that dominates, since the SW and LW cloud forcings are nearly equal and opposite. For the lower atmosphere and surface, the change in stratospheric humidity generates the largest forcing, at $\sim -2.5$ W m$^{-2}$. The decrease in ozone itself directly produces $\sim 1$ W m$^{-2}$ for the troposphere, but $\sim -12$ W m$^{-2}$ at the top of the atmosphere. Most of the latter forcing is felt by the stratosphere. Though ozone does provide a modest greenhouse effect to the troposphere because of the absorption around 9.6 $\mu$m, the increase in SW flux at the tropopause

outweighs this.

    Prior to the GOE, methane may have provided a significant greenhouse effect. The loss of this greenhouse due to a drop in methane abundance is often cited as a principle cause of the Huronian glaciations. A typical case from the oxidation simulations of Wogan et al. (2022) has methane drop from $10^{-4}$ to $\sim 5 \times 10^{-6}$; an estimate of the net change in forcing from methane is then $\sim 3 - 4$ W m$^{-2}$ (Figure 5 of Byrne and Goldblatt, 2014b) and the resulting temperature drop is expected to be $\sim 3$ K

(Figure 4 of Byrne and Goldblatt, 2015). The removal of most methane after the GOE thus constitutes a source of cooling, however, this is partially offset by the appearance of the ozone layer, which we find provides a positive forcing of $1.5 - 2$ W m$^{-2}$ when the effect on stratospheric humidity is accounted for. Accounting for all positive feedbacks in the GCM, the net forcing due to ozone is even larger, raising the surface temperature by $\sim 3$ K. Thus the change in oxidation state may not be enough to explain those early snowball events. Of course, this is an imperfect comparison: we are comparing the response of

1-D climate model to methane (Byrne and Goldblatt, 2015) and the response of a 3-D model to ozone (this study); further, the constraints on methane abundance do allow for larger changes across the GOE (Zahnle et al., 2019; Sauterey et al., 2020).

    The zonal wind is also affected by the decrease in ozone. The stratospheric jets weaken due to the change in the horizontal temperature gradient, while the jets of the upper troposphere strengthen. The effects we see are consistent with those found in Kiehl and Boville (1988), though in that study the largest changes took place in the northern hemisphere, while our results are

dominated by the southern hemisphere. The difference between their study and ours is probably a result of the time averaging interval: we average our results over 30 full years, while theirs used only 240 days, and thus is subject to seasonal effects. The Hadley cells increase in strength at low ozone, consistent with the increase in tropospheric jet speed.

    Comparing our Temperature Control and Constant CO$_2$ simulations, we can isolate the changes that are primarily controlled by the stratospheric response to low ozone. At the same ozone level, these simulations have very similar stratospheres but

quite different tropospheres (see Figure 3). Changes to high clouds and LW cloud forcing are thus governed by the impact on the stratosphere, while low clouds and SW cloud forcing are naturally sensitive to the temperature structure of the lower atmosphere.

    Low ozone by itself is not enough to explain any of the dramatic climate variations seen near the GOE, such as snowball episodes, though it was not expected to. Rather, the change in ozone is one piece of the puzzle, which we have attempted to

quantify in this work. Our follow-up studies will investigate the climate of Earth around the GOE, accounting for the other changes in greenhouse gases with atmospheric oxidation in the context of a different solar constant and rotation rate.





*Code availability.* The Community Earth System Model (CESM) version 1.2.2 used here is available at https://www.cesm.ucar.edu/models/. The 1-D flux calculations were done using the Python library Climate Modelling and Diagnostics Toolkit (climt) at https://github.com/CliMT/climt. Both CAM5 (a module of CESM) and climt employ the RRTMG radiation code, documented at http://rtweb.aer.com/rrtm_frame.html.

Our analysis and plotting scripts for all figures are available at https://github.com/deitrr/DG_Pre-Cambrian_Ozone.

*Data availability.* The CAM5 model outputs, ozone input files, and other data used in making the figures, are available at FRDR (provide link when available).

*Author contributions.* R.D. set-up and ran the simulations, performed the analysis, wrote all analysis and plotting code, and wrote the majority of the article. C.G. supervised and guided the project, provided much of the interpretation, and helped write and revise the article.

*Competing interests.* No competing interests are present.

*Acknowledgements.* We thank Eric Wolf for guidance in running CESM, Brandon Smith for further guidance in running CESM and for providing CAM model output, and Daniel Garduño Ruiz for photochemical model output. We also thank Victoria McDonald, whose documentation and code helped immensely with running CESM and with our analysis. This research was enabled in part by support provided by the BC DRI Group and the Digital Research Alliance of Canada (alliancecan.ca). Financial support to R.D. was provided by the Natural

Sciences and Engineering Research Council of Canada (NSERC; Discovery Grant RGPIN-2018-05929) and the Canadian Space Agency (Grant 18FAVICB21).



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
