# Peer review of "Effects of Ozone Levels on Climate Through Earth History"

_EGUsphere, 2022_

## Author Response (AR1)

Dear Editor,

Below we have included our responses to the two reviewer for the manuscript. These are copied directly from the interactive discussion, but we have added this document for thoroughness. Thank you for overseeing the editing and review of our article.

Thank you,

Russell Deitrick

**Response to reviewer 1:**

We've copied the reviewer comments for clarity; our responses to each comment are in bold font.

Thank you to Professor Kasting for the encouraging, thoughtful, and constructive review. This has helped us improve the manuscript significantly. We address each of your comments below.
* * *
1. (l. 29) "Therefore, the history of ozone is intimately connected to the history of molecular oxygen. For the first ~ 2 billion years of Earth's history (the Hadean and Archean Eons), oxygen and ozone were present only in abundances of â² 10–7 parts-per-volume (Zahnle et al., 2006; Lyons et al., 2014; Catling and Zahnle, 2020)."

--This statement might be true in the troposphere, but it is not true in the stratosphere. In the stratosphere, $O_2$ can reach mixing ratios of 1.-e3 to 1.e-2, depending on the $CO_2$ abundance. See, e.g., J. F. Kasting, Science (1993).

**RESPONSE: An excellent point. We have reworded this sentence and added a following sentence describing the possibility of stratospheric O2 due to due CO2: "…tropospheric oxygen was present only in abundances of $\leq 10^{-7}$ parts-per-volume, insufficient for an ozone layer to form (Zahnle et al., 2006; Lyons et al., 2014; Catling and Zahnle, 2020). Stratospheric oxygen would have been more abundant in a high $CO_2$ atmosphere, but insufficient to form a substantive ozone layer (Kasting, 1993; Segura et al., 2007)."**

2. (l. 38) "A second rise of oxygen occurred at the end of the Proterozoic Eon and beginning of the Phanerozoic Eon, bringing oxygen up to its modern value of ~ 21% (Lyons et al., 2014)."

--Well, maybe. Some authors (L.J. Alcott et al., Science, 2019) have argued that $O_2$ did not reach modern levels until around 400-450 Ma in what they term the 'Paleozoic Oxidation Event', or POE.

**RESPONSE: This is a good point. Our previous statement was overly simplified. We have reworded this as: "A second rise of oxygen occurred at the end of the Proterozoic Eon and beginning of the Phanerozoic Eon, bringing oxygen up to percent levels (Lyons et al., 2014).**

**Finally, recent work suggests an additional Paleozoic Oxidation Event that brought oxygen to approximately modern levels around 20% (Krause et al., 2018; Alcott et al., 2019). "**

3. (l. 44) "A key take-away from these studies is that ozone abundances are non-linear in oxygen concentrations—for example, ozone reaches near modern levels even at $O_2$ levels of â² $10^{-3}$ (Garduño Ruiz et al., 2022)."

--Is this an $O_2$ mixing ratio or is it a value in PAL (times the Present Atmospheric Level)? Also, not all models do this. I would reword it as: "for example, in some models (e.g., Garduño Ruiz et al., 2022) ozone reaches near modern levels even at $O_2$ levels of â² $10^{-3}$." It is a bit of a problem here that the Garduño Ruiz et al. paper is marked as 'submitted', so it is not possible to check these results.

**RESPONSE: Excellent suggestion, and yes, this was not clearly worded. This value is referring to the mixing ratio (ppv). We have reworded this phrase as follows: "For example, in some models (e.g., Garduno Ruiz et al., 2023), ozone reaches near modern levels even at $O_2$ mixing ratios of $\leq 10^{-3}$". Note also that the GR paper has just been published and is available here: https://www.sciencedirect.com/science/article/pii/ S0012821X23000845?dgcid=author**

4. (l. 50) "Such extremes have been principally studied using 1-D models (Morss and Kuhn, 1978; Levine and Boughner, 1979; Visconti, 1982; Kasting, 1987; Francois and Gerard, 1988). These works found that the global mean surface temperatures decreased by 5-7 K when ozone was removed."

--I didn't look at all these papers, but I looked at my own (Kasting, 1987). I got 3 degrees of warming from ozone and another 2 degrees from $O_2$ (mostly through pressure broadening of $CO_2$ and $H_2O$). Surface pressure was allowed to vary with $pO_2$ in that calculation. So, the sentence as written is not quite accurate. Looking forward at Figure 2 of this paper, it appears that the present authors also get about 3 or 4 degrees of warming from ozone. So, our results are not that different.

**RESPONSE: Apologies for misquoting the result from Kasting 1987–I think I overlooked the subtlety here, as you pointed out. We have corrected this and the value quoted in Section 4 to 3 K (the new text is below under point 8), and explain that molecular oxygen can contribute an additional 2 K through pressure broadening. A new footnote in Section 4 reads: "More specifically, Kasting (1987) found a total of 5 K difference in surface temperature due to a large change in oxygen. Of that, 3 K was the result of the radiative effect of ozone, while 2 K was the result of the pressure broadening by the presence of oxygen in high abundance (see Section 3.2)"**

5. (l. 94) "For ozone, we use vertical profiles from the photo-chemical calculations of Garduño Ruiz et al. (2022)."

--Well, again, this is a bit of a problem. What do these vertical profiles look like? It's really hard to review this manuscript rigorously without seeing the GR et al. paper. ..Ah, I got down further and saw that these profiles are shown in Fig. 1. Good! You should say that earlier.

**RESPONSE: That is a good idea. We have added a note at the end of this sentence referring to Figure 1, so that is clearer: "For ozone abundances, we use vertical profiles from the photo-chemical calculations of Garduno Ruiz et al. (2023); see Figure 1."**

6. (l. 324) "great oxidation event" should probably be capitalized.

**RESPONSE: Thanks for pointing this out. We have capitalized all instances of "Great Oxidation Event" in the revised manuscript.**

7. (l. 328) "While Kasting (1987) and Francois and Gerard (1988) found cooler surface temperatures, using 1-D radiative-convective models, the 3-D GCM modelling of Jenkins (1995, 1999) resulted in a warmer global mean surface temperature, using the GENESIS model."

--There is a known (to some) flaw in the GENESIS model calculations. See Payne, R. C., A. V. Britt, H. Chen, J. F. Kasting, and D. C. Catling (2016), The response of Phanerozoic surface temperature to variations in atmospheric oxygen concentration, J. Geophys. Res. Atmos., 121, 10,089–10,096. This paper was written in response to the following one: Poulsen, C. J., C. R. Tabor, and J. D. White (2015), Long-term climate forcing by atmospheric oxygen concentrations, Science, 348, 1238–1241. Poulsen et al. used the GENESIS model and found that decreasing $O_2$ increased surface temperatures. Payne et al. found the opposite using a 1-D climate model. The Poulsen et al. result is thought to have been caused by an unrealistic cloud feedback in GENESIS (D. Pollard, private communication). Dave Pollard, who worked here at Penn State for many years, was the chief architect of the GENESIS model.

**RESPONSE: Thank you for this information. We were not aware of this discussion outside the literature. Since neither of us (the authors) have experience with the GENESIS model, we cannot say directly whether the difference between our work and Jenkins shares the same cause as the difference between Payne et al and Poulsen et al. We have added a citation to Payne et al. (and some related works) to Section 3.2 where we discuss the pressure broadening effect, as it is relevant there. At the same time, we don't want to get into the potential issues with GENESIS or Poulsen et al., as those results are tangential to our study and discussion of that work would likely be a distraction. Here is the addition to Section 3.2: "Note, however, that changing oxygen can have a significant effect on climate by augmenting pressure broadening if the change is large enough to affect the total pressure (Kasting, 1987; Goldblatt et al., 2009; Payne et al., 2016; Wade et al., 2019). As previously stated, we have held the surface pressure constant in all our calculations, so this effect is ignored."**

8. (l. 330) "Now, with CESM 1.2.2, our results (for Constant CO2) are more similar to the Kasting (1987) and Francois and Gerard (1988) results, albeit with a smaller global-mean surface temperature difference: a cooling of ~5 K, compared to their ~ 5 K and ~ 7 K, respectively."

--As noted in point 4 above, the Kasting (1987) model predicted 3 degrees of cooling from removing ozone.

**RESPONSE: Thanks for pointing this out as well. We have corrected this to now read: "Now, with CESM 1.2.2, our results (for Constant CO2) are more similar to the Kasting**

**(1987) and Francois and Gerard (1988) results, albeit with a smaller global-mean surface temperature difference: a cooling of ~ 3.5 K, compared to their ~ 3 K  and ~ 7 K, respectively."**

**Response to reviewer 2:**

We've copied the reviewer's comments for clarity; our responses are in bold.

Thank you to the reviewer for the thoughtful and constructive comments. These will improve the manuscript immensely. Our responses to each individual comment are below.
* * *
L30 : specify whether the oxygen content corresponds to values for the troposphere.

**RESPONSE: The atmospheric abundances discussed here refer to the "base of the troposphere" (Catling & Zahnle 2020). We have rephrased this: "... tropospheric oxygen was present only in abundances of ≤ 10–7 parts-per-volume,"**

L32 : add capital letters to « great oxydation event »

**RESPONSE: Thank you for pointing this out. We have capitalized all instances of "Great Oxidation Event" in the revised manuscript.**

L35 : explain why the ozone layer is lower.

**RESPONSE: Good point, we should explain this. We've added a footnote explaining: "At lower oxygen levels, the peak ozone occurs at lower altitudes because the self-shielding effect of molecular oxygen is weaker. This allows the UV photons that photolyze oxygen (ultimately giving rise to ozone) to penetrate further into the atmosphere, to lower altitudes. Thus the peak in ozone production also occurs lower."**

L83-91 : Using an atmosphere model with a slab ocean considerably does not requires long integration time, but is the absence of a dynamic ocean not detrimental? Experiments performed with a coupled ocean-atmosphere model would permit to validate the approach here

**RESPONSE:  This is a good question, and a difficult one to answer. We simply don't have the computational resources to integrate all our simulations to equilibrium with a dynamic ocean as it typically takes thousands of years for the ocean to reach steady state. For this reason, it is an almost universal practice to use a slab ocean for deep paleoclimate. The few deep paleoclimate studies that include a dynamic ocean either do not run to full equilibrium or run only a small number of simulations. Perhaps in the next 5-10 years it will become feasible to integrate Earth system models for thousands of years for a large number of simulations, but for now, it is just not possible.**

L93 : The vertical profiles of oxygen and ozone are from the photo-chemical calculations of Garduño Ruiz et al described in a paper which is submitted at the time of submission of this paper. The authors do not explain how these profiles were obtained and the criteria that led them to choose these different profiles specifically.

**RESPONSE: Thanks for this point. Apologies that this was unclear. We have explained in more detail in the revised manuscript and moved this explanation to its own paragraph (as the current paragraph was getting a bit overly complicated). "The full text is now: "For ozone abundances, we use vertical profiles from the photo-chemical calculations of Garduno Ruiz et al. (2023); see Figure 1. Using the ATMOS model that originated with Kasting et al. (1979) and was updated by Arney et al. (2016) and Wogan et al. (2022), Garduno Ruiz et al. (2023) varied surface temperature and surface fluxes of oxygen and methane and ran the photo-chemistry to equilibrium. From these, we selected two pre-GOE profiles that were approximately an order-of-magnitude apart in surface $O_2$ abundance, and three post-GOE profiles that were also about an order-of-magnitude apart." Note also that the GR paper has just been published and is available here: https://www.sciencedirect.com/science/article/pii/S0012821X23000845?dgcid=author**

L109 : « In the Temperature Control sequence, we adjust the CO2 level to achieve a roughly constant mean surface temperature ». Does it mean in term of global mean annual temperature?

**RESPONSE: Thank you for the point of clarification. Yes, we do indeed mean the global-, annual-mean surface temperature. We have revised the text as: "...we adjust the $CO_2$ level to achieve a roughly constant globally and annually averaged surface temperature…"**

L128-130 : The authors suggest to use the EIS parameter. The authors should provide the equation (based on two papers) for the calculation of this parameter (and also the LTS parameter). Specify why these 2 parameters are only calculated over the ocean.

**RESPONSE: We have added the formula for LTS, since it is very simple, and have added the equation number for the LCL in Lawrence 2005. However, given that the EIS and LCL equations are clearly presented elsewhere, that we identify them by equation number in those works, and that these quantities are not critical to our conclusions, we are confident that our brief description is sufficiently clear. The full revised text is: "The LTS is simply the difference in potential temperature between the 700 hPa level and the surface (i.e., LTS = $\theta_{700} - \theta_0$). For the EIS, we use Equation 4 of Wood and Bretherton (2006). Their formula is dependent on the height of the lifting condensation level (LCL); for this we use the approximation given in Equation 24 of Lawrence (2005)." In addition, all our code for post-processing and plotting is available on our GitHub, so these calculations can be readily replicated by the reader. For ease of reference, the GitHub repository is here: https://github.com/deitrr/DG_Pre-Cambrian_Ozone**

**The reviewer is correct that it was unclear why we applied these only to marine clouds. We have added a clarifying statement: "The LTS and EIS are averaged over ocean regions only, as these parameters apply primarily to low-lying marine clouds."**

L173 : « The former roughly represents the tropopause »: Can we still speak of a tropopause if we refer to certain temperature profiles in Figure 3.

**RESPONSE: While the tropopause is often thought of as the temperature minimum above which the temperature begins to increase with altitude, it is better defined as simply the boundary between the troposphere (where convection can occur) and stratosphere (where convection is strongly inhibited). Practically, the tropopause can be diagnosed as the lowest model level with a zero net radiative flux. Even when the stratosphere is isothermal, as is the case with very low ozone, it is still stable against convection. So in that sense, an isothermal stratosphere is still a "stratosphere", i.e., it is stably stratified. These definitions (of troposphere, tropopause, and stratosphere) are typically used in deep paleoclimate studies wherein the vertical structure may be very different from modern day (Kasting & Donahue 1980, Kasting 1988, Zahnle, Claire & Catling 2006, for example).**

L323 : "great oxidation event" should be capitalized.

**RESPONSE: Thank you again. We've capitalized all instances in the manuscript.**

« with a focus on the climate before and after the great oxidation event ». The authors should rephrase the sentence because the boundary conditions (solar constant, paleogeography, pCO2, Earth's rotation rate...) are very different at GOE.

**RESPONSE: This is a fair point, since we haven't modeled all aspect of the GOE. We have rephrased this as "with a focus on ozone levels before and after the Great Oxidation Event."**

L341 : « it is unlikely that the complete removal of ozone would result in a temperature change greater than this ». Could the consideration of more realistic boundary conditions challenge this statement ?

**RESPONSE: We can see that our statement here was unclear. We have reworded this to align with our intended meaning: "A further decrease in surface temperature with ozone is unlikely, given the similarity of the simulations with nominal O2 = $10^{-7}$ and $10^{-9}$."**

L353 : « 10–4 » : typo

**RESPONSE: Thanks for catching that one. We have corrected this to "$10^{-4}$" as intended.**

L360 : « we are comparing the response of 1-D climate model to methane (Byrne and Goldblatt, 2015) and the response of a 3-D model to ozone ». Can the state of the Earth's surface (e.g. ice cover vs ice free Earth) modify the impact of oxygen?

**RESPONSE: This is a very good question and one we have not attempted yet to answer! Regarding ice cover specifically, it is difficult to say. There are known temperature effects, however, which will of course correlate with ice coverage (and make it difficult to tease out the effects of ice itself). For example, the temperature affects the chemical reactions that relate oxygen, ozone, and methane, thus a cold atmosphere with a certain abundance of**

oxygen or methane will have different abundances of ozone than a warm planet with the same oxygen or methane. So it is difficult to give a simple answer but this is a very interesting avenue for future efforts.